# Stress in Emergency Healthcare Professionals: The Stress Factors and Manifestations Scale

**DOI:** 10.3390/ijerph19074342

**Published:** 2022-04-05

**Authors:** Ángel García-Tudela, Agustín Javier Simonelli-Muñoz, José Miguel Rivera-Caravaca, María Isabel Fortea, Lucas Simón-Sánchez, María Teresa Rodríguez González-Moro, José Miguel Rodríguez González-Moro, Diana Jiménez-Rodríguez, Juana Inés Gallego-Gómez

**Affiliations:** 1Department of Urgency and Emergency, Hospital Clínico Universitario Virgen de la Arrixaca, 30120 Murcia, Spain; angelgt94@gmail.com (Á.G.-T.); lucasovera@hotmail.com (L.S.-S.); 2Faculty of Health Sciences, Catholic University of Murcia (UCAM), 30107 Murcia, Spain; mifortea@ucam.edu (M.I.F.); mtrodriguez@ucam.edu (M.T.R.G.-M.); jigallego@ucam.edu (J.I.G.-G.); 3Department of Nursing, Physiotherapy, and Medicine, University of Almeria, 04120 Almeria, Spain; djr239@ual.es; 4Department of Cardiology, Instituto Murciano de Investigación Biosanitaria (IMIB-Arrixaca), CIBERCV, Hospital Clínico Universitario Virgen de la Arrixaca, University of Murcia, 30120 Murcia, Spain; 5Department of Pneumology, Hospital Universitario Príncipe de Asturias, Alcalá de Henares, 28805 Madrid, Spain; respirama@yahoo.es

**Keywords:** nursing, stress, anxiety, optimism, emergency department professionals

## Abstract

Background: Healthcare workers are continuously exposed to a high level of stress, especially emergency department professionals. In the present research, we aimed to determine the internal consistency and validity of the Stress Factors and Manifestations Scale for in-hospital and out-of-hospital emergency workers. Methods: A quantitative, prospective, cross-sectional, and observational study including 269 emergency service professionals. Results: The scale was composed of 21 items, with a Cronbach’s α value of 0.908. The hospital workers (38.4 ± 10.8 vs. 35.1 ± 9.9, *p* = 0.014) and women (39.3 ± 11.4 vs. 34.2 ± 8.6, *p* < 0.001) had higher levels of stress. The sensitivity, specificity, and predictive values of the scale were adequate. Conclusion: In the present study, including in-hospital and out-of-hospital emergency workers, the Stress Factors and Manifestations Scale presented appropriate usefulness, internal consistency, and validity, with optimal predictive ability. Higher levels of anxiety, female gender, being less optimistic, and working in hospital emergency departments were related to increased stress levels. Further studies are warranted to validate our results and potentially extend the Stress Factors and Manifestations Scale to other contexts.

## 1. Introduction

Stress is defined as an imbalance in the body’s ability to cope with external demands [1]. Healthcare workers are continuously exposed to a high level of stress, especially emergency department professionals, due to the responsibility of their work and other stressors [2,3]. These highly demanding jobs, in turn, lead to increased dissatisfaction with work [4,5], negative health consequences [4], exhaustion [5], and poorer quality of sleep [6]. In turn, stress induces anxiety, defined as a feeling of nervousness, restlessness, anguish, and sometimes loss of control that can lead to physical and psychological disorders, which are commonly found in this type of service [7,8]. If this stress situation is maintained over a long period, occupational burnout syndrome may arise, and emergency professionals are more likely to develop it due to the conditions mentioned above [9].

This demanding work environment, associated with acute stress, may have negative consequences on patient health and safety, as it is associated with deficiencies in the performance of complex cognitive tasks and a greater number of healthcare errors [3,8,10]. 

This situation has been exacerbated by the appearance of the COVID-19 pandemic, which has had a great psychological impact on healthcare workers due to the adaptation of their jobs, isolation from their families, and fear of contagion due to their proximity to infected patients [11,12]. This has affected mental health and quality of life, especially for professionals who have dealt with the virus on the front line, such as emergency workers, as they have shown higher stress levels than other workers [13].

All of these drawbacks are also influenced by the degree of optimism of the individual, as there is a beneficial relationship between optimism and stress [14].

The prevalence of stress in emergency department professionals is high [15]. After taking stock of the above, the importance of having user-friendly tools to discover the level of stress of these workers becomes clear, as the results could be used to promote strategies to improve their work performance and quality of life and reduce errors in their practice due to lack of attention.

There are different tools for measuring the manifestations of stress. However, they tend to focus on a single domain, whereas this scale addresses life events, work events, and recognized symptoms of stress as a whole, so we believe that it can help other researchers to measure stress more accurately in healthcare workers and other groups [16,17,18,19].

Therefore, the present study aimed to determine the internal consistency and validity of the Stress Factors and Manifestations Scale for in-hospital and out-of-hospital emergency workers and to investigate whether a high level of stress is associated with anxiety, personal and work variables, and being less optimistic.

## 2. Materials and Methods

### 2.1. Study Design and Participants

We designed a study with a quantitative, prospective, descriptive, and cross-sectional methodology involving 269 in-hospital and out-of-hospital emergency department workers (Emergency Medical Ambulance) from the Region of Murcia (Spain). All the workers were recruited during their workday, from April 2019 to January 2020. They were recruited based on a consecutive non-probabilistic sampling procedure. The inclusion criteria were: workers who had been on sick leave due to a stressful event according to DSM-5 [20] diagnostic criteria or in treatment for a high level of stress and had worked in the emergency department over the past year. The participants were told about the purpose of the study and signed the informed consent form. 

### 2.2. Data Collected

Data were collected and processed anonymously. The study was authorized by the appropriate agencies as well as the ethics committee (Reference CE111707). It was conducted according to the standards set forth in the 1964 Declaration of Helsinki and its subsequent revisions. 

The data collected included personal and work issues information. Optimism about work was also measured, with a 5-point Likert scale [21], with the highest level of optimism scored with a 5. The self-made Stress Factors and Manifestations Scale (SFMS) was used to collect information on stress. A panel of 6 voluntary experts was utilized for the initial design of this scale. These experts were be registered nurses, psychiatrists, and psychologists with at least 5 years of experience in mental health. All of them were interviewed and informed individually about the study. The items composing the scale were obtained according to the main symptoms (anxiety, irritability, palpitations, sweating, etc.) and factors influencing stress (environment, work, life events, coping with problems) according to the scientific literature [22,23]. The final version of the self-reported scale included 21 items. As the study was conducted on in-hospital and out-of-hospital emergency department workers in Spain, the scale was designed in Spanish and afterward translated into English for the present manuscript. The translation was performed by an English native speaker with experience translating scientific texts. Importantly, the English version of the Stress Factors and Manifestations Scale is an exact translation of the content included in the original Spanish version (Appendix A).

The 21 items of the scale included 5-point Likert scores (from 1 = not at all to 5 = completely) and described different factors and manifestations of stress. The maximum scale score was 105 points, and higher scores indicated greater perceived stress. 

To measure anxiety, the State-Trait Anxiety Inventory (STAI) was utilized. This scale comprised 40 items and was designed to assess two independent concepts of anxiety, anxiety as a state (a transitory emotional condition) and anxiety as a trait (a relatively stable anxiety propensity). The time period for state anxiety is “right now, at this moment” (20 items) and for trait anxiety is “in general, most of the time” (20 items). Each subscale is composed of 20 items, answered with Likert-type responses. In this case, the responses were according to intensity (0 = rarely/not at all, 1 = somewhat/sometimes, 2 = quite often, 3 = very much/almost always). A higher score indicated a higher degree of anxiety [24].

### 2.3. Statistical Analysis

The software Ene 2.0 (GlaxoSmithKline, Brentford, UK) was used to calculate the sample size based on an estimation of 10.03 (SD) of stress obtained from other studies [25] with an accuracy of ±5%, an α error of 2%, and for an infinite population. A minimum sample of 96 emergency healthcare professionals was required.

Frequencies and percentages were used for the statistical analysis of the categorical values. On their part, median and interquartile ranges were used for continuous variables, although the mean ± standard deviation was used in the cases in which the distribution was normal according to the Kolmogorov-Smirnov test. The statistical test utilized to analyze the reliability of the scale was the test-retest. As for the Stress Factors and Manifestations Scale, a Cronbach’s α test was performed to measure its homogeneity or internal consistency, with a coefficient ≥0.70 defined as an ideal value. 

The individual analysis of each item was performed with the Homogeneity Index, evaluated with Pearson’s correlation coefficient. Each item with a coefficient >0.30 was considered useful for evaluating the attribute, excluding the items that did not meet this condition. 

The underlying dimensions present in the test were analyzed with a multivariate Factor Analysis statistical test. Before this analysis, to analyze construct validity, a factor analysis of the scale was performed to verify that it met the necessary criteria to be utilized. The appropriateness of the data was assessed with the Kaiser-Meyer-Olkin Index. The contrast of the correlation matrix was checked with Bartlett’s sphericity test.

A factor analysis was performed to explore the main components of the correlation matrix of all the scale’s items, with a Varimax rotation and the Kaiser’s criterion. Only the factors with values greater than 1 were extracted, as these explained the greatest percentage of the total variability. For this, the components extracted had to account for at least 60% of the variance explained in the correlation matrix. For the weights of the factor to be consistent, the criterion for an item to be part of the extracted factor was established as having a value greater than or equal to 0.40 [25].

To analyze convergent validity, the STAI questionnaire, used to measure state anxiety and trait anxiety, was taken as a reference and used in the study.

To contrast the association between the variables, Pearson’s chi-square test, Student’s t-test, and Pearson’s correlation were used. 

A value of *p* < 0.05 was accepted as statistically significant. Statistical analyses were performed using SPSS 21.0 for Windows (SPSS, Inc., Chicago, IL, USA).

## 3. Results

We included 269 emergency department workers from the Region of Murcia, with a mean age of 41.5 ± 10.7 years (59.5% women). The vast majority were married or living with a partner (61.3%), followed by single workers (26.8%), with the rest divorced or separated (11.9%). The professional categories were distributed as follows: 41.6% nurses, 24.9% physicians, 19% nursing assistants, 8.9% emergency technicians, and 5.6% resident physicians. Of these, 65.4% worked in the hospital’s emergency department. The mean time worked at the emergency department was 10.08 ± 8.7 years, and in the profession 15.2 ± 9.6 years. 32.7% admitted to having stress (Table 1).

To analyze the reliability of the questionnaire, a pilot study was conducted in a sample of 103 professionals, using the test-retest technique by repeating the questionnaire 14–21 days after completing it for the first time. The Spearman-Brown’s coefficient was 0.192. In the analysis of the two tests, the internal consistency of the items included in the questionnaire according to Cronbach’s alpha was 0.911 for the original test and 0.915 for the retest.

During the homogeneity analysis, no item was excluded from the Stress Factors and Manifestations Scale, as all had a correlation coefficient with the total of the corrected scale >0.30, obtaining a Cronbach’s α value of 0.908. The scale, with a range of possible values between 21 and 105 points, presented a mean value of 37.2 ± 10.6 (95% CI, 36.0–38.5). 

A factorial analysis was performed to check the construct validity of the scale. This analysis indicated the presence of an underlying four-factor structure, according to the Kaiser-Meyer-Olkin criteria. Bartlett’s sphericity test was also performed. In the factor analysis, the combined factors explained 60.5% of the variance, with each factor obtaining a value >0.40. The homogeneity test results for each of the four factors indicated values for Cronbach’s α higher than 0.70, and none of them were eliminated, as they all had a correlation coefficient above 0.30.

After the Varimax rotation, Factor 1 included eleven items related to “Self-concept”, Factor 2 included seven items related to “Sociability”, Factor 3 was composed of four items measuring “Somatization”, and Factor 4 included three items related to “Disease symptoms” (Table 2). The items included in Factor 1 (items 1, 2, 5, 6, 9, 10, 14, 16, 1, and 21) were related to each worker’s personality self-concept. This factor obtained a Cronbach’s α of 0.892 (mean 21.19 ± 6.90) and explained 42.68% of the variance. The items from Factor 2 (items 4, 5, 7, 8, 12, 17, and 21) corresponded to the relationship with others, with a Cronbach’s α of 0.810 (mean of 11.84 ± 3.97); it explained 6.47% of the total variance. As for the elements in Factor 3 (i.e., items 8, 13, 19, and 20), they were associated with physical manifestations of problems felt by emergency workers as a result of stressful situations. This factor obtained a Cronbach’s α of 0.757 (mean 5.57 ± 1.99) and explained 6.06% of the variance. Lastly, Factor 4 was composed of items related to disease symptoms (items 3, 11, and 15). This factor obtained a Cronbach’s α of 0.734 (mean 3.58 ± 1.12) and explained 5.38% of the variance (Table 2).

To analyze the convergent validity of the criteria, the State-Trait Anxiety Inventory questionnaire was used as a reference and used in the study to measure state anxiety and trait anxiety [24]. Table 3 shows the results, where a strong and positive correlation with high statistical significance is observed between the stress scale and its four factors and the STAI questionnaire in its two dimensions (state and trait). The cross-loading of all factor items was adequate. Table 3 shows the correlation between the factors and the total scale.

In the analysis of the results of the Stress Factors and Manifestations Scale values and its different factors, depending on the personal variables, we found that women suffered significantly higher levels of stress compared to men (39.3 ± 11.4 vs. 34.2 ± 8.6; *p* < 0.001). Neither marital status nor age were associated with stress, except in Factor 4 “Symptoms of illness” for age, where older workers obtained higher scores (R = 0.205, *p* = 0.001) (Table 4).

In the analysis of variables of the work type and their relationship with the Stress Factors and Manifestations Scale, we observed that the resident physicians had the most stress (43.0 ± 10.8), except in Factor 4 “Symptoms of illness”, followed by nursing assistants (40.4 ± 11.3) and nurses (37.5 ± 10.4). The lowest level of stress was observed for emergency technicians (31.5 ± 7.5), followed by physicians (35.2 ± 10.2), resulting in these statistically significant differences (*p* = 0.001). It was also found that hospital emergency workers were the most stressed compared to out-of-hospital emergency workers (38.4 ± 10.8 vs. 35.1 ± 9.9; *p* = 0.014). Years spent in the emergency department and years in the profession did not significantly affect stress, with only Factor 4 (disease symptoms) being statistically significant (R = 0.150, *p* = 0.014; R = 0.146, *p* = 0.016). Regarding optimism, the workers who were the most optimistic were those who had the least stress, as they obtained a negative correlation between optimism and all factors and, therefore, a significantly higher score in the total result of the scale (R = −0.351; *p* < 0.001) (Table 4).

Finally, we tested the predictive ability of the Stress Factors and Manifestations Scale for identifying stress in our population. Thus, the scale showed an excellent predictive performance with a c-index of 0.956 (95% CI 0.924–0.977, *p* < 0.001). The cut-off point of the scale for considering responders as having stress was 39, which presented the best combination of sensitivity (88.64%) and specificity (90.61%). 

## 4. Discussion

The main finding of the present study is the usefulness, reliability, and validity of the Stress Factors and Manifestations Scale tool to identify stressors and manifestations of stress in emergency department professionals. When analyzing the scale and performing the homogeneity analysis of the stress measurement instrument, we found a Cronbach’s α value of 0.908, a very significant value, considering that 0.70 is the minimum required. Therefore, this research demonstrates that the Stress Factors and Manifestations Scale is a validated scale for the study population and is very easy to use, where high scores strongly correlate with the level of stress. The present study also demonstrates that the Stress Factors and Manifestations Scale is a useful tool that could help health professionals and managers identify the main variables that affect stress in this specific population.

In addition, we found that the Stress Factors and Manifestations Scale converges with an internationally recognized and used anxiety measurement scale, the State-Trait Anxiety Inventory, used with all types of populations [24].

In our research, a strong and positive correlation (state anxiety: R = 0.693; trait anxiety: R = 0.765), with a high statistical significance (*p* < 0.001) was observed, between the stress scale and the State-Trait Anxiety Inventory questionnaire, coinciding this correlation with two other investigations. In one [26] conducted with emergency physicians, anxiety and stress were strongly associated, with more anxious individuals having higher stress levels. In the other [27], peak salivary cortisol response was significantly associated with higher STAI scores. This indicates that the existence of stressors in the emergency department staff, measured with the stress scale, implies a higher probability of anxiety states, both state and trait anxiety, thus indicating validity.

It is important to note that our results also revealed that out-of-hospital emergency workers had lower stress levels than in-hospital workers, with a marked statistical significance (*p* = 0.014), a finding of great value due to the absence of studies comparing such populations.

All the professional categories that make up both types of services participated in our investigation, with emergency technicians followed by physicians obtaining a lower score on the stress scale questionnaire and medical residents followed by nursing assistants obtaining the highest scores. Another article [16] showed that more than half the staff had stress, with no significant differences between the different professional categories.

On the other hand, research studies on hospital emergency departments are more numerous. Some studies were performed only on emergency nurses [28], and others on several professional categories [15], and the general results indicated the presence of medium-high stress levels. We believe that the differences between in-hospital and out-of-hospital emergency department stress levels could be due to the difference in working hours and poor sleep quality of these workers [29].

Another important finding obtained in the present study was the very varied results observed, in which notable levels of stress predominated in these professionals. However, no study has investigated and compared in-hospital and out-of-hospital emergency workers simultaneously. Almost no study has covered all the different professional categories that make up these services, making it difficult to compare the results measured with different scales and methods, thus making the present work pioneering research at the international level.

With respect to gender, we found that the results followed the same line as most studies, with women showing the highest stress levels [3]. On the other hand, in our study, neither age nor time spent in the emergency department and the profession were determinants for the levels of stress (except for the factor “Symptoms of disease” with which it is related), with research showing a disparity of opinion regarding the influence of these factors [3].

Lastly, another relevant aspect of our study was the view obtained about stress from the perspective of optimism. Although some articles in other types of populations have studied the influence of optimism, relating it to coping with stressful situations [14], in the present study we have identified a strong relationship between optimism and stress (R = −0.351, *p* < 0.001), and have found that the most optimistic workers had lower levels of stress, with statistically significant values. 

### 4.1. Limitations

Regarding the study’s limitations, this is a preliminary validation with preliminary data, so it would be interesting to validate this scale with workers from other hospitals or primary care services. The identified dimensionality of the scale has not been replicated in other independent samples with confirmatory factor analysis, nor was interobserver reliability assessed. Furthermore, it would be convenient to carry out this research in the health services from other cities to verify whether differences in the results exist.

### 4.2. Implications for Nursing Practice

Stress generates physiological, emotional, behavioral reactions, and negatively affects organizations. It may even cause burnout as a response to chronic stress. Thus, stress is of increasing interest in our daily lives, given its potential consequences on both employee health and business results (employment leave, absenteeism, and poor performance). 

Similarly, stress can exceed the capabilities of the individual and have negative consequences for in-hospital and out-of-hospital emergency workers. Hence, it can lead to excessive costs associated with these consequences and thus become a problem for workers, hospitals, and the health care system in general, with low productivity and poorer job satisfaction.

For the above reasons, identifying potential stressors and measuring stress is central to the appropriate management of emergency workers’ mental health. 

In short, we have created a concise and useful tool to detect stress as a response for in-hospital and out-of-hospital emergency workers. The novel Stress Factors and Manifestations Scale presented in this study is a simple and user-friendly tool that reliably assesses stress.

Therefore, we believe that the use of this tool is essential for the nursing profession. Once stress has been identified, strategies can be implemented to reduce stress and thus promote the better performance of these workers from the nursing management in the emergency department.

## 5. Conclusions

In the present study, including in-hospital and out-of-hospital emergency workers, the Stress Factors and Manifestations Scale presented appropriate usefulness, internal consistency, and validity, with optimal predictive ability. Higher levels of anxiety, female sex, being less optimistic, and working in hospital emergency departments were related to increased levels of stress. Further studies are warranted to validate our results and potentially extend the Stress Factors and Manifestations Scale to other contexts. 

## Figures and Tables

**Table 1 ijerph-19-04342-t001:** Descriptive variables.

Variables	Mean ± SD
**Age (years)**	41.5 ± 10.7
**Average time worked:**	
Emergency department	10.08 ± 8.7
Profession	15.2 ± 9.6
**Optimism about work**	3.85 ± 0.9
	N (%)
**Sex**	
Male	109 (40.5)
Female	160 (59.5)
**Marital status**	
Married	165 (61.3)
Single	72 (26.8)
Divorced/separated	32 (11.9)
**Professional categories**	
Nursing	112(41.6)
Medicine	67 (24.9)
Assistant	51 (19)
Emergency technician	24 (8.9)
Medical residence	15 (5.6)
**Emergency service**	
Out-of-hospital	93 (34.6)
In-Hospital	176 (65.4)
Stress	88 (32.7)

**Table 2 ijerph-19-04342-t002:** Factorial analysis of the questionnaire: matrix of rotated components.

Kaiser-Meyer-Olkin Index Bartlett’s Sphericity Test	0.917 <0.001			
Items	Factor 1 Self-Concept	Factor 2 Sociability	Factor 3 Somatization	Factor 4 Disease Symptoms
1. I feel restless	0.784			
2. I am anxious	0.730			
3. I feel sweating in some parts of my body				0.635
4. I have a negative attitude against others		0.670		
5. I feel disgusted	0.433	0.505		
6. I feel aggressive	0.662			
7. I have difficulties interacting with others		0.711		
8. I sleep worse than usual		0.432	0.600	
9. The work is beyond me or overtaking me	0.443			
10. I don’t eat the same as before	0.524			
11. I am not healthy enough to go to work				0.770
12. I tend to distrust people		0.687		
13. I have digestive problems			0.404	
14. I feel insecure	0.683			
15. I feel itching in some areas of my body				0.505
16. I feel overwhelmed	0.710			
17. I do not attend to my social relationships		0.666		
18. I postpone things for later	0.489			
19. I have muscle pain or tension			0.567	
20. I have heart palpitations			0.553	
21. I usually have doubts	0.606	0.433		
Self-values Variance	8.121 42.68%	1.425 6.47%	1.334 6.06%	1.184 5.38%

**Table 3 ijerph-19-04342-t003:** Analysis of the validity of the questionnaire’s criteria and the correlation between factors and the total scale.

	Factor 1 Self-Concept	Factor 2 Sociability	Factor 3 Somatization	Factor 4 Disease Symptoms	Total
STAI State	R: 0.697 *p* < 0.001	R: 0.560 *p* < 0.001	R: 0.476 *p* < 0.001	R: 0.315 *p* < 0.001	R: 0.693 *p* < 0.001
STAI Trait	R: 0.754 *p* < 0.001	R: 0.652 *p* < 0.001	R: 0.510 *p* < 0.001	R: 0.383 *p* < 0.001	R: 0.765 *p* < 0.001
Factor 1 Self-concept	1	R: 0.766 *p* < 0.001	R: 0.566 *p* < 0.001	R: 0.440 *p* < 0.001	R: 0.957 *p* < 0.001
Factor 2 Sociability	R: 0.766 *p* < 0.001	1	R: 0.521 *p* < 0.001	R: 0.434 *p* < 0.001	R: 0.871 *p* < 0.001
Factor 3 Somatization	R: 0.566 *p* < 0.001	R: 0.521 *p* < 0.001	1	R: 0.411 *p* < 0.001	R: 0.699 *p* < 0.001
Factor 4 Disease symptoms	R: 0.440 *p* < 0.001	R: 0.434 *p* < 0.001	R: 0.411 *p* < 0.001	1	R: 0.564 *p* < 0.001
Total	R: 0.957 *p* < 0.001	R: 0.871 *p* < 0.001	R: 0.699 *p* < 0.001	R: 0.564 *p* < 0.001	1

STAI: State-Trait Anxiety Inventory; R: Pearson’s correlation coefficient; *p*: statistical significance.

**Table 4 ijerph-19-04342-t004:** Association between the Stress Factors and Manifestations Scale and personal, job, and work optimism variables.

	Factor 1 Self-Concept	Factor 2 Sociability	Factor 3 Somatization	Factor 4 Disease Symptoms	Total
Personal variables					
Sex	19.0 ± 5.7	11.2 ± 3.4	5.0 ± 1.4	3.4 ± 0.9	34.2 ± 8.6
Male (n = 109)	22.6 ± 7.2	12.2 ± 4.2	5.9 ± 2.2	3.7 ± 1.2	39.3 ± 11.4
Female (n = 160)	*p* < 0.001	*p* = 0.031	*p* < 0.001	*p* = 0.031	*p* < 0.001
Marital status	21.4 ± 7.3	11.9 ± 4.3	5.4 ± 1.7	3.3 ± 0.8	37.4 ± 10.8
Single (n = 72)	20.7 ± 6.6	11.6 ± 3.7	5.4 ± 1.9	3.6 ± 1.2	36.6 ± 10.4
Married (n = 165)	23.0 ± 7.1	12.8 ± 4.1	6.1 ± 2.5	3.8 ± 1.2	40.5 ± 11.2
Divorced/Separated (n = 32)	*p* = 0.203	*p* = 0.248	*p* = 0.207	*p* = 0.130	*p* = 0.169
Age	R = −0.027	R = 0.105	R = 0.082	R = 0.205	R = 0.051
	*p* = 0.663	*p* = 0.085	*p* = 0.181	*p* = 0.001	*p* = 0.404
Work variables					
Professional category	21.2 ± 6.4	11.9 ± 4.0	5.8 ± 2.2	3.4 ± 0.9	37.5 ± 10.4
Nursing (n = 112)	19.8 ± 6.7	11.4 ± 3.7	4.9 ± 1.3	3.6 ± 1.3	35.2 ± 10.2
Medicine (n = 67)	23.1 ± 7.3	12.2 ± 4.1	6.0 ± 2.2	3.9 ± 1.3	40.4 ± 11.3
Assistant (n = 51)	17.5 ± 5.2	10.4 ± 2.9	4.5 ± 0.7	3.3 ± 0.8	31.5 ± 7.5
Emergency technician (n = 24)	26.0 ± 7.7	13.6 ± 4.2	5.8 ± 1.7	3.1 ± 0.3	43.0 ± 10.8
Medical residence (n = 15)	*p* < 0.001	*p* = 0.107	*p* = 0.001	*p* = 0.033	*p* = 0.001
Emergency service	19.8 ± 6.4	11.2 ± 3.5	5.1 ± 1.5	3.5 ± 1.2	35.1 ± 9.9
Out-of-hospital (n = 93)	21.9 ± 7.0	12.1 ± 4.1	5.7 ± 2.1	3.5 ± 1.0	38.4 ± 10.8
In-Hospital (n = 176)	*p* = 0.018	*p* = 0.095	*p* = 0.006	*p* = 0.975	*p* = 0.014
Years in emergencies	R = −0.055	R = 0.071	R = −0.039	R = 0.150	R = −0.005
	*p* = 0.370	*p* = 0.248	*p* = 0.526	*p* = 0.014	*p* = 0.941
Years in profession	R = −0.039	R = 0.102	R = 0.043	R = 0.146	R = 0.029
	*p* = 0.522	*p* = 0.094	*p* = 0.487	*p* = 0.016	*p* = 0.634
Optimism about work	R = −0.352	R = −0.297	R = −0.165	R = −0.267	R = −0.351
	*p* < 0.001	*p* < 0.001	*p* = 0.008	*p* < 0.001	*p* < 0.001

R: Pearson’s correlation coefficient *p*: statistical significance.

## Data Availability

The data presented in this study are available on request from the corresponding author.

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
