# Peer review of "Stress in Emergency Healthcare Professionals: The Stress Factors and Manifestations Scale"

_ijerph, 2022, doi:10.3390/ijerph19074342_

Round 1

Reviewer 1 Report

The authors can deeply benefit from the evidence regarding healthcare workers' outcomes during the first wave of the covid-19 pandemic in the following and very recently published work to strengthen their literature background.

I suggest searching through the recommended systematic revision using "healthcare" as a keyword since it is a very long one. However, it presented a lot of research about stress, anxiety, depression, and coping strategies of healthcare workers that should be accounted for. 

Line 80. Your claim should be expanded providing a clear picture about what were the identified main factors influencing stress according to the expert panel. 

The sample size is not justified and this is a major issue from my point of view. Please justify why 269 is sufficient for conducting your analyses, both factor analysis, and validity-related analysis. 

Please provide a reference justifying the retainment of items with loadings higher than 0.40. 

I did not find a clear statement about possible items' cross-loadings across factors. Did the authors consider this issue? Intercorrelation among factors is not presented either. Since the authors computed a total score for their scale (Table 2) I think the factors are correlated to one another and I would like to see evidence of that. 

Overall, I would suggest the authors smooth their claims about the fact that they "demonstrated" the usefulness, internal consistency, and validity of their scale. From my point of view, this is a preliminary validation with preliminary data. The identified dimensionality of the scale has not been replicated in other independent samples with confirmatory factor analysis.  Moreover, as long as the authors do not provide evidence of adequate power for their analyses, I cannot rule out the fact that the study is underpowered. 

Line 210 is written in Spanish. 

The limitations section is too short and unaware of the true limits of the work which I hope after this review of mine will be clearer.

Author Response

Reviewer 1

The authors can deeply benefit from the evidence regarding healthcare workers' outcomes during the first wave of the covid-19 pandemic in the following and very recently published work to strengthen their literature background.

>>> Thank you for the comment. We have added information on covid-19.

I suggest searching through the recommended systematic revision using "healthcare" as a keyword since it is a very long one. However, it presented a lot of research about stress, anxiety, depression, and coping strategies of healthcare workers that should be accounted for.

>>> We apologize but seems that the reviewer forgot to explicitly cite the reference of the recommended systematic review. However, we have reinforced our work with other studies regarding this issue. If you provide us with the mentioned reference, we will be happy to include it.

Line 80. Your claim should be expanded providing a clear picture about what were the identified main factors influencing stress according to the expert panel.

>>> We have added the information in the text.

The sample size is not justified and this is a major issue from my point of view. Please justify why 269 is sufficient for conducting your analyses, both factor analysis, and validity-related analysis.

>>> We apologize for this. The Sample size calculation has been added in the revised version of the manuscript.

Please provide a reference justifying the retainment of items with loadings higher than 0.40.

>>> We have added the reference in the text.

I did not find a clear statement about possible items' cross-loadings across factors. Did the authors consider this issue? Intercorrelation among factors is not presented either. Since the authors computed a total score for their scale (Table 2) I think the factors are correlated to one another and I would like to see evidence of that.

>>> A statement on cross-loadings has been added to the manuscript. In addition, the correlation values between the individual factors and the total scale have been added.

Overall, I would suggest the authors smooth their claims about the fact that they "demonstrated" the usefulness, internal consistency, and validity of their scale. From my point of view, this is a preliminary validation with preliminary data. The identified dimensionality of the scale has not been replicated in other independent samples with confirmatory factor analysis. Moreover, as long as the authors do not provide evidence of adequate power for their analyses, I cannot rule out the fact that the study is underpowered.

>>>We have added the changes to the text.

Line 210 is written in Spanish.

>>> We are really sorry for this mistake. We have changed the sentence as appropriate.

The limitations section is too short and unaware of the true limits of the work which I hope after this review of mine will be clearer.

>>> We have expanded the limitations section in the revised manuscript. Thank you.

Reviewer 2 Report

Thank you for giving me the opportunity to review the manuscript. I cannot accept this manuscript for publication in the current form.

1) In the introduction section, the novelty of this assessment scales in the population is not always clear. There have been a variety of scales to assess stress, but the authors anly showed that "There are deferent tools to measure the manifestations of stress, however, all have a specific focus on other populations from those of our study." Please show these tools more specifically and clearly. The information the authors showed was too limited to find the novelty of this study.

First, it is necessary to review these assessment scales.

Second, please explain what is the difference between the previous stress scales and this scale in views of the PECO. 

Third, please show what is the novelty and the advantage/disadvantage of this scale, compared with the previous scales.

2) The inclusion criteria and exclusion criteria of the participants were too ambiguous. The authors only showed "The inclusion criteria were: not being diagnosed with any stress-related illness and having worked in the emer- 
gency department at least in the last year. The participants were told about the purpose of the study and signed the informed consent form. " What is the definion of "any stress-related illness". What kind of diagnostic criteria were selected??

3) What is "a Likert scale"? Please refer to the previous study to assess the validity and reliability of this scale.

4) What is the difference between "the perceived stress scale (PSS)" and "Stress Factors and Manifestations Scale (SFMS)"?? Why the authors seleced the SFMS in this study instead of PSS in spite of the fact that both scales assessed perceived stress??

5) Please show how to assess inter-rater reliability, test-retest reliability and internal consistency reliability of SFMS, and the results of them.

6) Please attach the STARD 2015 checklist and fill in the page numbers.

7) Whether participants formed a consecutive, random or convenience serieshether participants formed a consecutive, random or convenience series??

8) Please define what is the index test and reference standard in detail to allow replication.

9) Please show the definition of and rationale for test positivity cut-offs or result categories of both the index test and the reference standard, distinguishing pre-specified from exploratory. Please explain the construct validity in detail, including convergent validity, known-groups validity, and structual validity of the outcome scales including SFMS.

10) Please show whether clinical information and reference standard results were available to the performers/readers/assessors of the index test.

11) Please show estimates of diagnostic accuracy and their precision (such as 95% confidence intervals).

12) Please show any adverse events from performing the index test or the reference standard.

13)  I think that it is impossible to make a conclusion that "the usefulness, internal consistency, and validity of the Stress Factors and Manifestations 
Scale for in-hospital and out-of-hospital emergency workers have been demonstrated." Please change this sentence to show clearly the limitation of the study and what kind of the study are warranted.

I cannot accept this manuscript for publication without revising the manuscript for the above reasons.

Author Response

Reviewer 2

Thank you for giving me the opportunity to review the manuscript. I cannot accept this manuscript for publication in the current form.

1) In the introduction section, the novelty of this assessment scales in the population is not always clear. There have been a variety of scales to assess stress, but the authors anly showed that "There are deferent tools to measure the manifestations of stress, however, all have a specific focus on other populations from those of our study." Please show these tools more specifically and clearly. The information the authors showed was too limited to find the novelty of this study.

>>> The text has been modified in a summarized form, and the requested data are explained in more detail below.

First, it is necessary to review these assessment scales.

>>> Some of the most widely used are the following:

Perceived Stress Scale (PSS) (1983) was first used to assess stressful situations in college students. It measures the degree to which individuals perceive life situations in the last month as stressful. It presents 3 versions (14, 10 and 4 items) with questions referring to the last month.

Depression, Anxiety and Stress Scale (DASS) (1995) was developed to evaluate symptoms of depression, anxiety and stress. It presents 2 versions (42 and 21 items) with questions referring to the last month.

Stress Reactivity Index (IRE) (1981). 32 items. Shows habitual reactions in stressful situations.

Social Read-justment Ratin Scale (SRRS) (1967). 41 items. Measures the most frequent life events, but does not focus on the work environment, with questions referring to the last year.

In the health care field we can highlight The Nursing Stress Scale (NSS) (1981). 34 items. It measures the frequency with which certain situations are perceived as stressful by the nursing staff.

Second, please explain what is the difference between the previous stress scales and this scale in views of the PECO. 

>>> None of the scales mentioned above is adapted to our study setting because of the following:

Depression, Anxiety and Stress Scale (DASS) analyzes dimensions other than stress.

Stress Reactivity Index (IRE) focuses only on habitual stress reactions and is extensive.

Social Read-justmentRatin Scale (SRRS) focuses on the most frequent life events, but not on the work environment. It is also very long and covers a very long period of time.

The Nursing Stress Scale (NSS) is focused on situations related to dealing with patients, doctors and other colleagues without taking into account other situations that may cause stress in daily life or paying attention to other symptoms that the person may suffer, and it would not be valid for other health categories.

And finally, the Perceived Stress Scale (PSS) is perhaps the most widely used and useful scale and could be adapted to measure stress in general, but it does not include any item on the work environment or on symptoms described in patients with stress.

Third, please show what is the novelty and the advantage/disadvantage of this scale, compared with the previous scales.

>>> The stress of emergency workers has always been high, and these figures have increased with the appearance of the COVID-19, so we consider the great usefulness of this tool, which has proven to be useful in this population and could be extrapolated to other sectors. Our scale is characterized by being an easy-to-use instrument, which can be self-administered, brief, and which in turn is more complete than other scales for measuring stress of great reputation, but whose creation dates back decades. We believe that science has updated the knowledge we have of stress since the creation of the most widely used scales on this subject, for this reason we believe that we must also update these measurement tools, to adapt them as much as possible to the current situation.  With our scale we manage to measure stress in a more updated and accurate way, especially because it deals with items on life events, work and recognized symptoms of stress, unlike the most commonly used scales, so we believe that it can help other researchers to measure stress in health workers and other groups.

2) The inclusion criteria and exclusion criteria of the participants were too ambiguous. The authors only showed "The inclusion criteria were: not being diagnosed with any stress-related illness and having worked in the emer- gency department at least in the last year. The participants were told about the purpose of the study and signed the informed consent form. " What is the definion of "any stress-related illness". What kind of diagnostic criteria were selected??

>>> We apologize if this was not clear in our previous version. By "any stress-related illness" we refered to workers who had been on sick leave due to a stressful event according to DSM-5 diagnostic criteria or under treatment for a high level of stress, since this situation could condition their answers in the questionnaire.

It has been modified in the text.

3) What is "a Likert scale"? Please refer to the previous study to assess the validity and reliability of this scale.

>>> Likert scales are psychometric instruments where respondents must indicate their agreement or disagreement on a statement, item by item, which is done through an ordered and unidimensional scale.

We have added the reference in the text.

4) What is the difference between "the perceived stress scale (PSS)" and "Stress Factors and Manifestations Scale (SFMS)"?? Why the authors seleced the SFMS in this study instead of PSS in spite of the fact that both scales assessed perceived stress??

>>> The Perceived Stress Scale (PSS), unlike the scale of Stress Factors and Manifestations (SFMS), does not include any item on the work environment or on symptoms described in patients with stress, which are two very important factors that influence stress, so we consider it to be much more complete for measuring stress in workers.

5) Please show how to assess inter-rater reliability, test-retest reliability and internal consistency reliability of SFMS, and the results of them.

>>> Inter-observer reliability was not assessed. This will be reported in the study limitations.

Test-retest reliability results are added in the manuscript. The internal consistency reliability is described in the results section, with the Cronbach's alpha value.

6) Please attach the STARD 2015 checklist and fill in the page numbers.

>>> It has been attached to the consignment.

7) Whether participants formed a consecutive, random or convenience serieshether participants formed a consecutive, random or convenience series??

>>>This information has been included in the manuscript.

8) Please define what is the index test and reference standard in detail to allow replication.

>>> We have added the information in the text.

9) Please show the definition of and rationale for test positivity cut-offs or result categories of both the index test and the reference standard, distinguishing pre-specified from exploratory. Please explain the construct validity in detail, including convergent validity, known-groups validity, and structual validity of the outcome scales including SFMS.

>>> Thanks for this suggestion. We have now provided the cut-off value of the Stress Factors and Manifestations Scale. We have also added some text regarding the construct validity in the manuscript.

10) Please show whether clinical information and reference standard results were available to the performers/readers/assessors of the index test.

>>> We have added the information in the text.

11) Please show estimates of diagnostic accuracy and their precision (such as 95% confidence intervals).

>>> We really thank the reviewer for this interesting suggestion. In this revised version of the manuscript we have included the predictive accuracy and (95% CI) of the Stress Factors and Manifestations Scale, according to the c-index.

12) Please show any adverse events from performing the index test or the reference standard.

>>> We have not found any.

13)  I think that it is impossible to make a conclusion that "the usefulness, internal consistency, and validity of the Stress Factors and Manifestations 
Scale for in-hospital and out-of-hospital emergency workers have been demonstrated." Please change this sentence to show clearly the limitation of the study and what kind of the study are warranted.

>>> Thank you, we have modified the sentence.

Round 2

Reviewer 1 Report

Dear authors, thank you for modifying the paper accordingly to my suggestions. I would like to point out that I did not forget the explicitly cite the reference since it was removed by the editorial office. Of course, the disagreement between us should not impact your revision process, thus I will not penalize your paper for not mentioning the "phantom" work.

Reviewer 2 Report

I think that this manuscript would be suitable for publication.